# The impact of music education on children's cognitive and socioemotional development: A quasi-experimental study in the Guri Program in Brazil

Graziela Bortz[1]*, Beatriz Ilari[2], Nayana Di Giuseppe Germano[3],
Andrea Parolin Jackowski[4,5‡], Hugo Cogo-Moreira[6‡], Patrícia Silva Lúcio[7]

**1** Department of Music, Unesp, São Paulo, São Paulo, Brazil, **2** Thornton School of Music, USC, Los Angeles, California, United States of America, **3** Department of Music, Federal University of Santa Maria, Santa Maria, Rio Grande do Su, Brazil, **4** Department of Health & Wellness Design, School of Public Health, Indiana University, Bloomington, Indiana, United States of America, **5** Department of Psychiatry, Federal University of São Paulo, São Paulo, SP, Brazil, **6** Department of Education, Østfold University College, Østfold, Fredrikstad, Norway, **7** Department of Psychology and Psychoanalysis, State University of Londrina, Londrina, Paraná, Brazil

☯ These authors contributed equally to this work.
‡ APJ and HC-M also contributed equally to this work.
* graziela.bortz@unesp.br

## Abstract

### Background

Transferability of music education to cognitive and social skills has recently been explored, but its causal effects remain debatable.

### Objectives

To examine the impact of the Guri music education program on children's cognitive and socioemotional skills in children from underserved communities in São Paulo, Brazil.

### Design

#### A quasi-experimental study

**Participants.** Children at risk for behavioral problems who participated in the Guri Program (n = 37, 5–8 years) and matched controls from the same community (n = 61, 6–7 years).

### Intervention

The children attended music classes once a week for seven months. Follow-up (FUP) was carried out after seven months of classes.

**Data availability statement:** We have a restriction imposed by the Consent Form signed by both researchers and parents that affirms: "The researchers are committed to using the data collected only for this research." In case it is strictly necessary to check the data, they are available from the Ethical Committee Plataforma Brasil n. 15049919.8.0000.5420 at https://plataformabrasil.saude.gov.br.

**Funding:** Fapesp Public Policies Research Program (Proc. n. 2019/02133-4). Fundunesp (Proc. n. 3334/2022 – PIF).

**Competing interests:** The authors have declared that no competing interests exist.

## Main outcomes

Behavioral problems (measured by SDQ) and other secondary measures (the Digit-span subtest from WISC-IV, PARB-Q questionnaire, and attention test, BPA). Control measures: ABEP SES questionnaire and Raven's matrices? test.

## Statistical analysis

Baseline comparisons were performed using t-tests for independent samples and chi-squared tests, and the intervention effect was analyzed using generalized linear models (GLM). For multiple comparisons, Bonferroni correction was applied for the p-value cutoff. SPSS (version 30.0) was used for data analysis.

## Results

Since the groups differed in terms of maternal education at baseline, these data were used as a control variable in the GLM analysis. The group effect was not significant for behavioral or cognitive measures at the FUP, suggesting that the intervention did not produce relevant outcomes. Behavioral problems and alternating attention at baseline significantly predicted the respective measures at FUP, indicating a developmental effect.

## Conclusions

Although they did not reach statistical significance, the total difficulties in the SDQ and emotional symptoms may be potentially relevant behavioral aspects for future research.

## Introduction

### Music education and executive functions

The hypothesis that music education affects cognitive abilities and brain plasticity, with potential transferability to other domains, has been the focus of several studies published over the last decade [1–11]. While some studies suggest that abilities related to executive functions (EFs) are enhanced by musical training [2–4], others remain inconclusive [11,12]. Nonetheless, inhibitory control, one of the central dimensions of EFs, appears consistently to benefit from music education/music training, especially among preschoolers [12].

EFs consist of a set of abilities that support goal-directed and autonomous behavior, which are essential for establishing daily goals, as well as more complex, long-term planning [13–17]. Although validated models of EFs remain a topic of debate [14,18,19], there is consensus among the proposed theoretical frameworks regarding key dimensions such as attention, inhibitory control, working memory (WM), fluid intelligence, problem solving, decision making, and cognitive flexibility.

EFs can be classified as *hot*, or those associated with emotional and social cognitive aspects relevant to decision-making process, or *cold*, which are linked to mental abilities involved in inhibitory control, WM, short-term memory (STM), reasoning and planning [20,21]. Cold EF tasks require mental effort and constant top-down supervision of the pre-frontal cortex (PFC) to prevent automatic actions [16].

According to Diamond's model [17], attention, a subdomain of inhibitory control, can be influenced by emotional and cognitive self-regulation [22,23], even though it belongs to EF cold domains. Because attention demands effort and volition, it can be involved in both hot and cold EFs depending on the type of stimulus to be inhibited (*e.g.*, self-control in delayed gratification or resistance to a distractor in attentional focus). The two types of attention (emotional and cognitive) may not fall under the same factor but are strongly correlated [17,24]. Petersen and Posner's [23] updated model of attention [22] outlines three major networks: alerting, orienting, and executive networks. The latter includes focal attention and regulation of processing networks.

Attention can also be classified according to the type of task: [1] sustained attention/focused (*i.e.*, maintaining focus on a single stimulus/task over time), [2] divided attention (*i.e.*, concentrating on two or more stimuli/tasks simultaneously for a given time), and [3] shifting/alternating attention (*i.e.*, changing the focus of attention to the perceptual attributes of the stimuli under relevant stimulus-response map rules) [25,26]. Attention also plays an important role in WM, another EF cold domain, as it requires processing, storage, and sharing of cognitive resources from both [27]. Therefore, it is necessary to maintain a focus of attention by inhibiting the preponderant stimulus to prioritize information, allowing updates of the environment, and subsequently coordinating them with automated responses in the use of WM. This occurs when new associations are created in the STM.

A meta-analysis by Talamini et al. [8] revealed a musician advantage for STM and WM with tonal stimuli, moderate effects with verbal stimuli, and nonsignificant effects with visuospatial stimuli. Similarly, a systematic review by Rodriguez-Gomez and Talero-Gutierrez [12], which examined the associations between WM and music training, found that three of eight studies reported improvements in either visual or spatial WM following music interventions in children. Additionally, four studies indicated improvements in selective attention among children, although only one study showed statistical significance. Furthermore, of the nine studies conducted with preschoolers, six demonstrated improvements in inhibitory control following music education. These findings suggest that verbal WM and selective attention may be promising outcomes of music education.

## Hot executive functions, social competence, and social skills

Hot EFs are far less studied than cold EFs, and despite the fact that psychometric instruments that access emotion self-regulation, decision-making, empathy, and delayed gratification are often used [15]. One reason for such limited interest may be the overlap between hot EFs, social competence, social skills, and self-regulation constructs. Zimmerman et al. [20], defined hot executive functions as "cognitive abilities supported by emotional awareness and social perception (*e.g.,* social cognition)." This concept was built on Chan et al. [28] and McDonald's models [29], separating social cognition from emotion cognition and decision-making, and positioning both under the umbrella of hot EFs. In turn, social cognition is divided into cold (*i.e.*, cognitive empathy and theory of mind) and hot (*i.e.*, affective empathy and theory of mind, and emotion recognition) aspects [20]. For example, while one person may make decisions based on *understanding* others' cues, others may arrive at decisions based on their *affections*. Unsurprisingly, both cold and hot social cognition have neural correlates [30–32].

Social competence refers to the ability to maintain positive relationships while pursuing personal goals [33]. For Hukkelberg et al. [34], whereas social competence is a broad term, social skills encompass more specific, observable behaviors, which compound a social competence factor [35–38]. Both constructs correlate positively with academic achievement, problem-solving skills [35,38,39], and mental health outcomes [35,38]. Social skills can be measured using scales that address cooperation, self-control, internalizing and externalizing problem behaviors, empathy, and hyperactivity [35,38,39].

## Social skills and music

The relationship between music education, emotional self-regulation, and prosocial skills remains unclear. Some studies have demonstrated positive effects of musical interventions on prosocial skills in preschoolers and school-aged children

[40–42], while others have found moderate to null effects [11,43,44]. Interestingly, some studies have shown improvements in specific social skills among children and youth. For example, Villanueva et al. [45] demonstrated enhancements in children's sharing behaviors, theory of mind, and state empathy following formal music instruction. Alemán et al. [11] found a significant improvement in behavioral difficulties detected via parental questionnaires and scales, especially among male children who had experienced violence and mothers with low levels of formal music education. Ilari et al. [46] found a significant reduction in children's aggressiveness and hyperactivity following four years of music education. Similarly, Boal-Palheiros and Ilari [38] found a reduction in externalizing behavioral problems among Portuguese children from underserved communities after one year of music education. Therefore, current evidence is mixed, mainly for children from non-WEIRD (i.e., non-White, English-speaking, industrialized, rich, and democratic) populations [47] and those at risk for behavioral problems.

**The present study.** This study aims to contribute to the developmental literature by examining the impact of music education on specific cognitive and socioemotional outcomes among children at risk of behavioral problems. This quasi-experimental investigation builds on existing cross-sectional and correlational research that found evidence supporting the beneficial effects of music education for these outcomes. We focused on aspects of attention (shifting, dividing, and sustained) and aggressive behavior, comparing test results from a group of children who participated in formal music classes with a control group from the same community.

As noted, many studies exploring the associations between music, EFs, and social skills have been conducted in the global North, predominantly in North America and Western Europe, and in WEIRD populations [47]. This study aimed to investigate whether participation in *Projeto Guri* [48] – a musical program designed to serve low-income communities in the city of São Paulo, Brazil, – reaps cognitive and social skills benefits for children, including those at risk for behavioral problems. (The data discussed here were first presented as a thesis of senior professor exam in 2023 by the corresponding author. It remains unpublished, and the results were reviewed in this article.)

## Method

### Ethics

All research protocols were approved by the local ethics committee *of Plataforma Brasil* n. 15049919.8.0000.5420, and the Rebec n. U1111-1243–8071. Written informed consent was obtained from the municipality of São Paulo, Brazil. A Memorandum of Understanding was created between the State University of São Paulo (Unesp) and the Guri Program, outlining the specifics of the research, expectations, and obligations of all the involved parts.

### Design

This was a longitudinal study with a quasi-experimental design. This design allowed us to examine the effects of children's participation in the Guri project on cognitive and socioemotional outcomes. Because quasi-experimental designs lack randomization, using control variables is key to mitigate biases caused by confounding variables that could influence a child's likelihood of participating in the Guri program (for further details on causality and the principle of ignorability, see [49]). Randomization was not feasible due to the ethical and democratic enrollment processes inherent to the music program. The children also underwent MRI scans at baseline and follow-up. These data are not included in this article. For adherence, the children then received a gift, a snack, a certificate of contribution to scientific research and their families received a refund for transportation.

### Sample and recruitment

Children and their families were invited to participate in the study within their music programs (Guri *polos*) and local schools. We conducted the intervention study in 2022 during the Brazilian academic year (February to December). The

original schedule proposed in the trial study protocol predicted that the intervention would take place during the 2020 academic year, preceded by a pilot study in 2019. Due to the Covid-19 pandemic, the intervention had to be interrupted during the initial data collection, in March 2020, and the study was resumed in 2022 with new participants. The researchers responsible for the statistical analyses were blinded to the participants and the examiners were unaware of the outcomes being examined.

The target music group consisted of children aged 5–9 years who were about to begin music classes for the first time in the Guri Program [48]. Some amendments were made to the original trial study protocol due to difficulties in participant enrollment. As mentioned earlier, the study began shortly after schools reopened in Brazil. Because the study involved MRI scans, we initially aimed to recruit children aged 6–7 years in a 2:1 ratio, targeting 50 children in the experimental group and 100 in the control group. Unfortunately, adherence to the recruitment was low, which we believe was partly due to culture changes following the pandemic period. Government records indicated higher levels of absenteeism and school dropouts, along with a decline in public school enrollment [50]. Families' interest in the Guri program also decreased significantly after this period, according to its managers. As a result, we extended the target age range and ultimately achieved an experimental-to-control group ratio of 5:3. This large Program consists of a network of music education centers, called *polos,* distributed across the metropolitan area of São Paulo, also known as *Grande São Paulo. Most polos* are located in the city's outskirts and operate within a public system of integrated schools called *Centros Educacionais Unificados* (Unified Educational Centers) or CEUs. Children in the music group were recruited from 10 different *polos*: seven from the eastern region of the city, which is also the most populated, three from the southern region, and two from the northern region. Music classes for beginners took place weekly for one hour and followed a modality known as *musicalização* [51], a general music approach involving collective singing, movement, performance of rhythms and melodies, and the use of small percussion instruments, recorders, and Orff xylophones and metallophones.

The control group consisted of children in the same age range who met the same inclusion criteria and attended regular schools near the *Guri polos* (see below). The exclusion criteria for both groups were as follows: 1) prior participation in regular school music classes or instrumental/voice lessons and 2) diagnosis of autism spectrum disorder (ASD) or Attention Deficit Hyperactivity Disorder (ADHD). Children in the experimental group were recruited during their enrollment in the Guri Program when parents/guardians were informed about the study and provided with consent forms. After signing consent forms, parents completed the Strengths and Difficulties Questionnaire (SDQ–parent version) [52,53] and the ABEP questionnaire [54] that assesses socioeconomic status (SES). This initial screening phase aimed to identify children at risk for behavioral problems in four of the five SDQ domains: emotional symptoms, conduct problems, hyperactivity, and relationship problems.

Recruitment in schools took place following authorization from the school boards. Researchers met with parents to explain the study, obtain consent, and distribute the questionnaires. When in-person meetings were not possible, questionnaires were sent home with children and subsequently collected. A total of 255 parent protocols were collected from the *Guri* Program and surrounding schools. Of these, 143 protocols were excluded from the SDQ cut-off. Additional screening using the Brazilian version of Raven's test [52,53] excluded 10 children identified as being at risk for intellectual difficulties. One child was excluded for having previous music lessons and another was diagnosed with autism. In total, there were 155 exclusions. The final sample consisted of 100 participants: 39 in the music group and 61 in the control group.

During the follow-up (FUP), two children were excluded from the music group due to a late autism report, resulting in 37 children in the music group. Baseline data were collected in March and April 2022, and FUP tests and questionnaires were administered seven months later. This study started at 2020, but when the intervention was about to begin, the peak of the coronavirus pandemic come. By that time, although the baseline assessment had already finished, the children of the experimental group were taking remote music classes and the research had to be cancelled. Therefore, this sample is different from that in their pre and FUP assessments.

At baseline, children in the music group were 5–8 years old (mean = 6.54; S.D. = 0.61) and 51.4% boys (n = 19). In the control group, the children were primarily boys (70.5%; n = 43) and 5–7 years old (mean = 6.28; S.D. = 0.49). Six children from the control group were not reassessed at FUP for logistic reasons (Fig 1). Children who dropped out did not differ from those who completed FUP in terms of sex ($\chi^2$ [1] = 0.890, $p$ = 1.000), group assignment ($\chi^2$ [1] = 0.199, $p$ = 0.229), or age ($t$(96) = 1.701, p = 0.092).

## Instruments

### Control variables

• *Maternal education*: We used the Brazilian questionnaire of economic classification criteria, *Associação Brasileira de Empresas de Pesquisa* [54], created to assess socioeconomic status based on household samples. This questionnaire contains nine items that assess a) possession of durable consumer goods, b) type of water supply system and street paving, c) number of residents in the household, d) family composition, and e) level of education of the family head. The results of the questionnaire were stratified into five socioeconomic classes: A (subdivided into A1 and A2), B (also subdivided into B1 and B2), C, D, and E. For this study, we used maternal education measured by years of schooling in this questionnaire.

• *Raven's Colored Progressive Matrices* (CPM)–Special Scale [55,56]. We used this test to assess non-verbal fluid intelligence, analogical reasoning, and the ability of participating children to generate responses from non-verbal stimuli. This test is composed of 36 dichotomously corrected items (score of 0 for incorrect answers and 1 for correct answers), with a total score of 36. Normative data are presented in percentiles and attributed to age groups. Children with nonverbal intelligence falling under percentile 25 were excluded from the sample.

### Subject variables

• *Strengths and Difficulties Questionnaire* (SDQ)–parent version [52,53] and Brazilian version [57]. SQD provides an assessment of the behavior, emotions, and relationships of children and adolescents. Questions were grouped into five dimensions: emotional symptoms, conduct problems, hyperactivity/inattention, peer relationship problems, and prosocial behavior. Each item is scored between 0 and 2, according to respondents' agreement with the statements (*e.g.*, if the parent thinks that the statement "Restless, overactive, cannot stay still for long" is certainly true about his/her child, the item is scored 2 for the hyperactivity-inattention dimension). Some items presented opposite directional relationships with the construct and had to be reversed before scoring. Each dimension is scored on a scale of 0–10. The higher the score, the higher the behavioral difficulty or prosocial behavior. Difficulty items were summed into a difficulty score ranging from 0 to 40. To select children at behavioral risk, we used cutoffs of difficulty scores ≥ 10, but the prosocial behavior scores varied across the sample.

• *Digit span subtest for children from the WISC-IV* [58,59]. In this test, children are presented with increasingly long strings of numbers, composed by 2–9 items each (*e.g.*, 2-9-7). Upon hearing the sequence, the child must repeat the digits in the same order (short-term memory task) or in inverse order (working memory task). Each sequence size was presented in two different trials, and the child received a score of 1 for each sequence correctly uttered. The total score for each task is 16. The test was discontinued when the child made an error in two trials with the same string of numbers. In this study, we used the span score in reverse order, or the longest sequence numbers kept by the child in memory, as this subtest has been recognized as a robust measure of working memory [60].

• *Peer Aggressive and Reactive Behaviors Questionnaire* (PARB-Q) [61], Brazilian version [62]. This self-report questionnaire measures behaviors in terms of active and reactive aggression among pairs [59]. This instrument contains

**Flow of participants through each stage of the trial.**

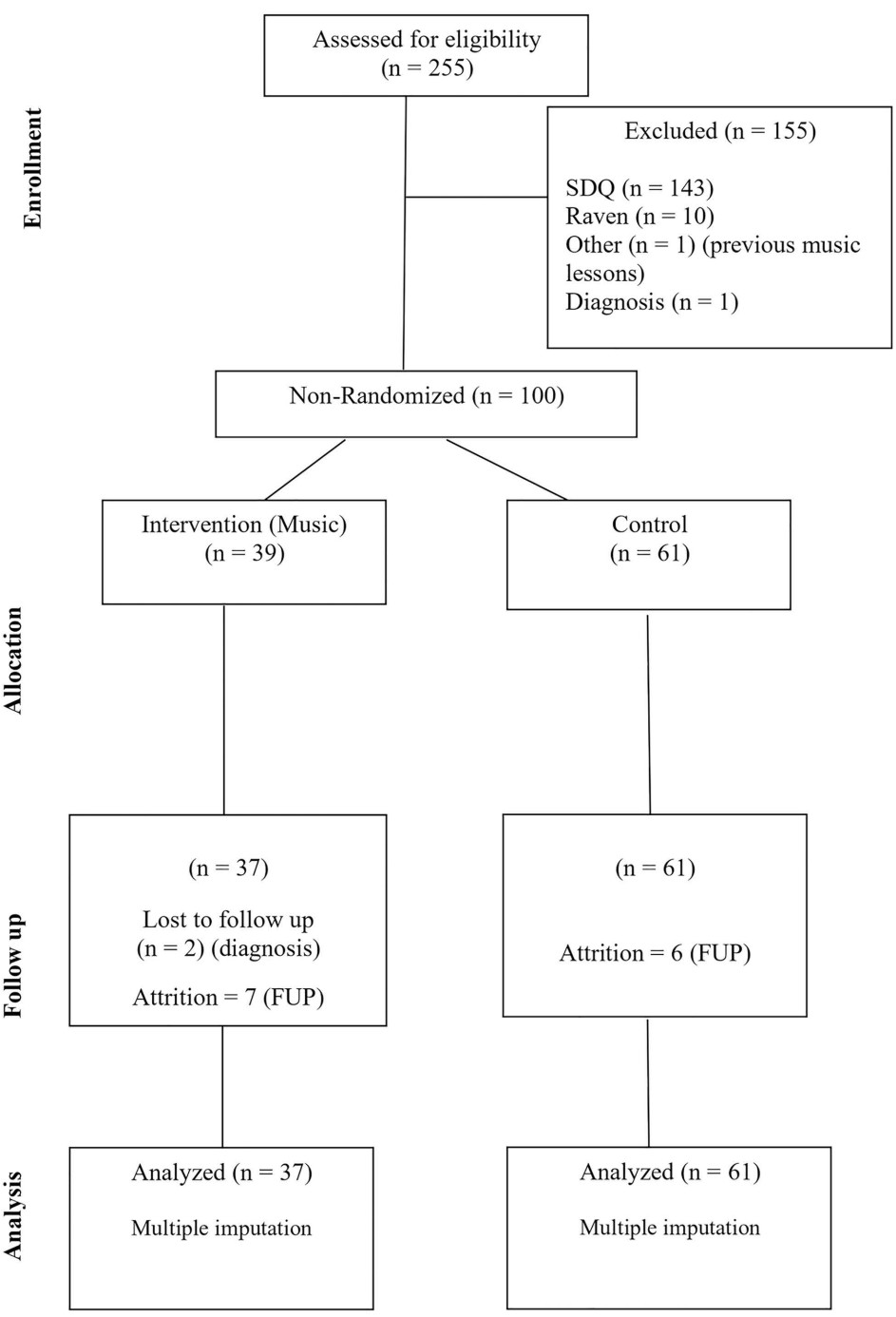

**Fig 1. Flowchart of the participants at each trial stage.**

20 questions and two scales: the Peer Aggression Scale (PA) and the Reaction to Peer Aggression Scale (RPA). PA consists of five items that assess proactive (or instrumental) physical and verbal aggression (*e.g.*, kicking or slapping) and three social-desirability bias items that are not scored (*e.g.*, telling jokes). The questions begin with "how often do you...," followed by the circumstances to be answered. The RPA consists of 12 sentences containing different ways children report how they react to aggression from their peers. Questions such as when a colleague of yours..., "when a colleague of yours…," are followed by statements, which may signal reactive aggression (RA; 6 items, *e.g.*, "hit or push you," "do you hit your colleague?"), seeking teacher support (STS; 3 items, *e.g.*, "hit or push you, do you tell the teacher?"), and internalized emotional reactions (IR; 3 items, *e.g.*, "hit or push you, do you cry and sulk?"). Each question was rated on a 4-point Likert scale (every day, sometimes, a few times, and never). PA scores range from 0 to 15, and RPA scores range from to 0–18 on the RA scale and from to 0–9 on the STS and IR scales.

- *Bateria Psicológica para Avaliação da Atenção* (BPA; Psychological Battery for Attention Assessment) [63]. The BPA is a pencil-and-paper test divided into three subtests that assess different domains of attention: sustained attention (in one object among distractors), divided attention (focus on three objects among distractors), and alternating attention (alternating focus between different objects among distractors).

The entire test battery was completed in 40–50 minutes, depending on each child's performance. The children were allowed to take breaks during the test sessions.

## Statistical analysis

Multiple imputations were performed to deal with missing data. The percentage missing for each individual variable varied from 1.02% (PARB-Q measures in the baseline) to 14.29 (SDQ and PARB-Q measures in the FUP). Ten imputations were performed using fully conditional specification as a custom method and linear regression as the model type for the scale variables.

Descriptive statistics (means and standard deviations) were available for the baseline and FUP measures. Pairwise comparisons between the groups at baseline were performed using *t*-tests for independent samples (metric variables) and chi-squared statistics (ordinal/nominal variables). Because SPSS does not provide pooled results for multiple imputed data, standard deviations, Cohen's *d* [64], and chi-square statistics were manually processed in Excel using Rubin's rule [65].

The intervention effect was analyzed using Generalized Estimating Equations (GEE), an approach suitable for modeling longitudinal or repeated-measures data, as it considers the dependence of observations within individuals and groups. Respondents were specified as the within-subject factor, and assessment time (baseline and FUP) was modeled as the repeated measure. An exchangeable working correlation structure was assumed. Group and assessment time were included as fixed factors, with maternal education entered as a covariate. Main effects were reported, and the group×time interaction was tested to assess the intervention's effectiveness.

Data was analyzed using SPSS 30.0 for Windows. Statistical significance was set at $p < 0.05$. Owing to multiple comparisons (15 pairwise comparisons in the baseline and 14 GLM analyses), we applied Bonferroni's correction to the *p* value, dividing 0.05 for the number of comparisons. Therefore, the new *p-values adopted were* 0.0033 and 0.0036 (bilateral). We reported the *t*-tests statistics according to Levene's Test for Equality of Variances results (equal variances were assumed if *p*-values were larger than 0.0033, otherwise equal variances were not assumed). Cohen´s *d* presents the following classification: 0.30 = low; 0.5 moderate; 0.80 high.

## Results

### Control variables

Table 1 presents the descriptive statistics and comparisons of the control variables. Children in the experimental group tended to be older, to present higher levels of nonverbal intelligence, and their mothers tended to be more educated.

**Table 1. Descriptive statistics of the control variables at baseline.**

| Variables | Experimental | Control | Levene´s test | Comparison |
|---|---|---|---|---|
| Age in baseline (years) | 6.54 (0.61) | 6.28 (0.49) | $Z = 6.685$, $p = 0.010$ | $t(96)=2.350$, $p = 0.021$, $d = 0.535$[a] |
| Sex (male) | 51.4% ($n = 19$) | 70.5% ($n = 43$) | – | $\chi^2$ [1] =3.631, $p = 0.057$ |
| Raven | | | | |
| Superior | 13.5% ($n = 5$) | 3.3% ($n = 2$) | | $\chi^2$ [3]=7.919, $p = 0.048$ |
| Medium superior | 32.4% ($n = 12$) | 18.0% ($n = 11$) | – | |
| Medium | 37.8% ($n = 14$) | 47.5% ($n = 29$) | | |
| Medium inferior | 16.2% ($n = 6$) | 31.1% ($n = 19$) | | |
| Maternal education[b] | | | | |
| Elementary school (illiterate/incomplete) | 2.7% ($n = 1$) | 15.0% ($n = 9$) | | $\chi^2$ [4]=28.930, $p < 0.001$[c] |
| High school (not complete) | 8.1% ($n = 3$) | 18.3% ($n = 11$) | – | |
| Undergraduate (not complete) | 32.4% ($n = 12$) | 58.3% ($n = 35$) | | |
| Undergraduate (complete) | 56.8% ($n = 21$) | 8.3% ($n = 5$) | | |

[a]Equal variances assumed; [b] There was one participant from the control group with missing data; [c]Results for pooled chi-squared; adjusted $p$-value was 0.0033.

Conversely, the control group was comprised of a larger number of boys. Because of Bonferroni's correction, the differences achieved significance only for maternal education, and this variable was entered as a covariate in further analysis.

## Behavioral and cognitive results

Table 2 presents the descriptive statistics for the sample at baseline ($n$, mean, and standard deviation).

*T*-tests for independent samples showed absence of differences between the groups in the baseline. Therefore, any of these variables were included as covariates in the GLM statistics (Table 3).

GEE models were used to verify the impact of musical intervention on the different variables of the study. For the SDQ analysis, difficulties (*i.e.*, hyperactivity, relationship problems, conduct problems, emotional symptoms, and difficulties in general) and prosocial behavior were considered as dependent variables. We considered the group (control, experimental) and time (baseline, FUP) as factors and maternal education as covariate. Due to the ordinal nature of this variable, dummy codes were created using the lowest level of the maternal education (illiterate) as the reference variable. The control group and baseline were used as references.

In the GEE analysis, the main effect of groups indicates whether the groups differ significantly on the target variable, regardless of time. Similarly, the main effect of time indicates whether scores on the target variable change (decrease or increase) regardless of group. These results are adjusted for maternal education. The effect of the intervention is reflected in the presence of a significant group × time interaction, which shows whether changes in the dependent variable (decrease or increase) differ between groups. We expect a greater reduction in symptoms or a greater increase in cognitive variables for the treatment group compared to the control group. Baseline and FUP comparisons are shown in Table 4.

As seen in Table 4, after adjusting for maternal education, no significant differences were found between groups at FUP in behavioral measures ($p > 0.0033$). However, a significant reduction in difficulties was observed over time ($p < 0.001$), reflecting developmental effects, with an average decrease of 3 points from baseline to FUP.

Comparisons between groups for direct aggressive behavior and reaction to aggression assessed by PARB are shown in Table 5. No significant effects were found for aggression measures.

The results for verbal working memory assessed by digit span (reverse order) and attentional measures are shown in Table 6. No significant effect was observed for the cognitive measures. Nevertheless, a significant increase in alternate attention was observed over time ($p < 0.001$), regardless of the group, with an average increase of 19 points from baseline to FUP.

**Table 2. Descriptive statistics of the baseline and the follow-up measures (*n*, means, and standard deviation).**

| Variables | Baseline | | | | Follow-up | | | |
|---|---|---|---|---|---|---|---|---|
| | Experimental | | Control | | Experimental | | Control | |
| **SDQ** | *n* | Mean (sd) | *n* | Mean (sd) | *n* | Mean (sd) | *n* | Mean (sd) |
| Difficulties (total score) | 37 | 15.32 (4.47) | 61 | 14.61 (4.28) | 30 | 11.90 (5.03) | 54 | 13.87 (6.98) |
| Hyperactivity/Inattention | 37 | 5.62 (2.20) | 61 | 5.33 (2.26) | 30 | 4.27 (2.24) | 54 | 4.53 (2.72) |
| Peer relationship problems | 37 | 2.54 (1.73) | 61 | 2.43 (1.64) | 30 | 2.07 (2.00) | 54 | 2.17 (1.85) |
| Conduct problems | 37 | 2.65 (1.60) | 60 | 2.65 (1.98) | 30 | 2.00 (1.37) | 54 | 2.89 (2.15) |
| Emotional Symptoms | 37 | 4.51 (2.46) | 61 | 4.51 (2.46) | 30 | 3.57 (2.10) | 54 | 4.23 (2.60) |
| Prosocial behavior (free variation) | 37 | 8.05 (1.60) | 61 | 8.18 (1.60) | 30 | 8.00 (1.84) | 54 | 8.28 (2.15) |
| PARB | | | | | | | | |
| Direct aggression | 37 | 13.30 (2.21) | 60 | 12.27 (2.86) | 30 | 13.80 (1.65) | 54 | 13.31 (2.26) |
| Reactive aggression | 37 | 15.68 (3.42) | 60 | 13.97 (4.31) | 30 | 16.40 (2.46) | 54 | 15.63 (3.63) |
| Internalizing behavior | 37 | 5.03 (3.10) | 60 | 4.38 (3.11) | 30 | 5.17 (2.97) | 54 | 4.48 (2.88) |
| Seeking teacher support | 37 | 2.65 (2.34) | 60 | 3.15 (3.39) | 30 | 2.23 (2.73) | 54 | 2.28 (2.26) |
| **WM** | | | | | | | | |
| Reverse span | 37 | 2.03 (0.99) | 61 | 1.59 (1.27) | 30 | 2.73 (0.91) | 55 | 2.40 (0.97) |
| **Attention** | | | | | | | | |
| Sustained Attention | 37 | 30.81 (26.68) | 61 | 22.62 (23.96) | 30 | 42.80 (21.49) | 55 | 34.67 (20.78) |
| Divided Attention | 37 | 15.78 (34.48) | 61 | 7.46 (39.57) | 30 | 34.27 (30.57) | 55 | 24.06 (38.85) |
| Alternating Attention | 37 | 28.76 (36.56) | 60 | 24.90 (17.08) | 30 | 47.17 (16.82) | 55 | 38.40 (16.59) |

PARB = Peer Aggressive and Reactive Behaviors Questionnaire. WM = working memory. Standard deviations are shown in parentheses.

**Table 3. Comparison between the groups in the subjects´ variables (*t*-tests).**

| | Levene´s test | t-test | Cohen´s *d* | Result |
|---|---|---|---|---|
| **SDQ** | | | | |
| Difficulties (total score) | Z = 0.380, p = 0.539[b] | t(96) = 0.791, p = 0.431 | 0.17 | NS |
| Hyperactivity/Inattention | Z = 0.189, p = 0.664[b] | t(96) = 0.630, p = 0.530 | 0.13 | NS |
| Peer relationship problems | Z = 0.382, p = 0.538[b] | t(96) = 0.328, p = 0.743 | 0.07 | NS |
| Conduct problems[a] | Z = 1.208, p = 0.274[b] | t(96) = −0.466, p = 0.659 | −0.09 | NS |
| Emotional Symptoms | Z = 1.463, p = 0.229[b] | t(96) = 1.053, p = 0.295 | 0.22 | NS |
| Prosocial behavior (free variation) | Z = 0.132, p = 0.718[b] | t(96) = −0.352, p = 0.726 | −0.07 | NS |
| PARB | | | | |
| Direct aggression[a] | Z = 4.590, p = 0.035[b] | t(96) = 1.830, p = 0.067 | 0.39 | NS |
| Reactive aggression[a] | Z = 4.937, p = 0.029[b] | t(96) = 2.192, p = 0.039 | 0.44 | NS |
| Internalizing behavior[a] | Z = 0.004, p = 0.950[b] | t(96) = 0.939, p = 0.348 | 0.20 | NS |
| Seeking teacher support[a] | Z = 9.615, p = 0.003[c] | t(804) = −0.903, p = 0.367 | −0.18 | NS |
| **WM** | | | | |
| Reverse span | Z = 14.431, p < 0.001[c] | t(96) = 1.790, p = 0.060 | 0.37 | NS |
| **Attention** | | | | |
| Sustained Attention | Z = 0.079, p = 0.079[b] | t(96) = 1.571, p = 0.065 | 0.33 | NS |
| Divided Attention | Z = 0.001, p = 0.0981[b] | t(96) = 1.058, p = 0.292 | 0.22 | NS |
| Alternating Attention[a] | Z = 5.246, p = 0.024[b] | t(96) = 0.894, p = 0.371 | 0.22 | NS |

[a]Results are based on polled statistics (Rubin´s rules). [b] Equal variances assumed; [c] Equal variances not assumed; Levene's test for original data; Adjusted *p*-value is 0.0033.

**Table 4. GEE results for behavioral measures (SDQ)[a].**

| Variables in the model | β | SD. Error | 95% CI Lower bound | 95% CI Upper bound | Sig. |
|---|---|---|---|---|---|
| Intercept (outcome: Difficulty) | 14.806 | 1.6609 | 11.545 | 18.067 | <0.001 |
| Main effects | | | | | |
| Group (ref. control) | −0.597 | 0.9116 | −2.384 | 1.190 | 0.513 |
| Time (ref. baseline) | **−3.003** | **0.8505** | **−4.686** | **−1.321** | **<0.001** |
| Maternal education (ref. illiterate) | | | | | |
| High school (not complete) | 0.945 | 2.1640 | −3.229 | 5.189 | 0.662 |
| Undergraduate (not complete) | 0.301 | 1.5912 | −2.822 | 3.424 | 0.850 |
| Undergraduate (complete) | 0.607 | 1.7466 | −2.2825 | 4.039 | 0.728 |
| Interaction (treatment effect) | | | | | |
| Group x time | 2.289 | 1.133 | 0.069 | 4.510 | 0.043 |
| Scale | 27.542 | 0.1761 | | | |
| Intercept (outcome: Hyperactivity) | 5.323 | 0.7851 | 3.783 | 6.864 | <0.001 |
| Main effects | | | | | |
| Group (ref. control) | 0.002 | 0.4616 | −0.903 | 0.907 | 0.996 |
| Time (ref. baseline) | −1.240 | 0.4609 | −2.145 | −0.335 | 0.007 |
| Maternal education (ref. illiterate) | | | | | |
| High school (not complete) | 0.241 | 0.8927 | −1.511 | 1.992 | 0.787 |
| Undergraduate (not complete) | −0.160 | 0.7487 | −1.629 | 1.309 | 0.831 |
| Undergraduate (complete) | 0.583 | 0.8025 | −0.992 | 2.158 | 0.468 |
| Interaction (treatment effect) | | | | | |
| Group x time | 0.503 | 0.5767 | −0.631 | 1.637 | 0.383 |
| Scale | 5.888 | 0.2973 | | | |
| Intercept (outcome: Relationship problems) | 1.883 | 0.4566 | 0.987 | 2.780 | <0.001 |
| Main effects | | | | | |
| Group (ref. control) | −0.163 | 0.3358 | −0.822 | 0.495 | 0.626 |
| Time (ref. baseline) | −0.386 | 0.3223 | −1.017 | 0.246 | 0.231 |
| Maternal education (ref. illiterate) | | | | | |
| High school (not complete) | 0.934 | 0.4536 | 0.045 | 1.824 | 0.040 |
| Undergraduate (not complete) | 0.837 | 0.4166 | 0.020 | 1.654 | 0.045 |
| Undergraduate (complete) | 0.546 | 0.5151 | −0.467 | 1.559 | 0.290 |
| Interaction (treatment effect) | | | | | |
| Group x time | 0.140 | 0.3869 | −0.619 | 0.898 | 0.718 |
| Scale | 3.028 | 0.0443 | | | |
| Intercept (outcome: Conduct problems) | 2.943 | 0.7960 | 1.371 | 4.514 | <0.001 |
| Main effect | | | | | |
| Group (ref. control) | 0.111 | 0.4096 | −0.692 | 0.914 | 0.786 |
| Time (ref. baseline) | −0.506 | 0.2892 | −1.074 | 0.062 | 0.081 |
| Maternal education (ref. illiterate) | | | | | |
| High school (not complete) | 0.087 | 0.8830 | −1.651 | 1.826 | 0.921 |
| Undergraduate (not complete) | −0.377 | 0.7257 | −1.809 | 1.055 | 0.604 |
| Undergraduate (complete) | −0.315 | 0.8473 | −1.991 | 1.360 | 0.710 |
| Interaction (treatment effect) | | | | | |
| Group x time | 0.563 | 0.4053 | −0.233 | 1.358 | 0.165 |
| Scale | 3.439 | 0.0480 | | | |
| Intercept (outcome: Emotional symptoms) | 4.670 | 0.7274 | 3.243 | 6.097 | <0.001 |

*(Continued)*

**Table 4.** (Continued)

| Variables in the model | β | SD. Error | 95% CI Lower bound | 95% CI Upper bound | Sig. |
|---|---|---|---|---|---|
| Main effect | | | | | |
| Group (ref. control) | −0.547 | 0.5735 | −1.671 | 0.577 | 0.340 |
| Time (ref. baseline) | −0.872 | 0.3605 | −1.581 | −0.163 | 0.016 |
| Maternal education (ref. illiterate) | | | | | |
| High school (not complete) | −0.332 | 0.7676 | −1.838 | 1.174 | 0.666 |
| Undergraduate (not complete) | −0.015 | 0.6054 | −1.204 | 1.174 | 0.981 |
| Undergraduate (complete) | −0.219 | 0.7274 | −1.647 | 1.208 | 0.763 |
| Interaction (treatment effect) | | | | | |
| Group x time | 1.083 | 0.4619 | 0.177 | 1.989 | 0.019 |
| Scale | 5.149 | 0.0462 | | | |
| Intercept (outcome: Prosocial behavior) | 8.293 | 0.6886 | 6.940 | 9.645 | <0.001 |
| Main effect | | | | | |
| Group (ref. control) | −0.140 | 0.3673 | −0.860 | 0.580 | 0.703 |
| Time (ref. baseline) | 0.030 | 0.3715 | −0.720 | 0.780 | 0.936 |
| Maternal education (ref. illiterate) | | | | | |
| High school (not complete) | −0.755 | 0.8688 | −2.460 | 0.951 | 0.385 |
| Undergraduate (not complete) | 0.368 | 0.6611 | −0.931 | 1.666 | 0.578 |
| Undergraduate (complete) | −0.523 | 0.7281 | −1.953 | 0.907 | 0.473 |
| Interaction (treatment effect) | | | | | |
| Group x time | 0.032 | 0.4257 | −0.813 | 0.876 | 0.941 |
| Scale | 3.496 | 0.2554 | | | |

[a]SDQ = Strengths and Difficulties Questionnaire.

## Discussion

The aim of this quasi-experimental study was to examine the impact of a music education program on aspects of attention—shifting, divided, and sustained—, socioemotional skills, and reactive and aggressive behaviors in children aged 5–8 from underserved communities of São Paulo. Using standardized tests, we compared results from a group of children who participated in a community-based music program to matched controls at two time points: before the intervention and seven months later. Because maternal education is known to influence children's cognitive and socioemotional development [66–68], controlled this variable through the Brazilian ABEP socioeconomic status questionnaire [54]. To our knowledge, this is likely the first quantitative study carried out on the impact of sociomusical programs in Brazil.

The main finding of our study is that the music intervention did not demonstrate an effect on cognitive and socioemotional skills. After controlling for maternal education, and applying Bonferroni's correction for multiple comparison, group membership did not impact the outcomes assessed at FUP. However, our study highlights relevant developmental issues for this age group by demonstrating that behavioral problems and attentional difficulties tend to remain stable over the observed period. In contrast, developmental changes were identified for total difficulties and alternating attention, with a decrease in total difficulties and an increase in attention. These findings suggest that screening these skills may be important for referrals to early intervention.

Our findings contradict earlier theoretical and empirical studies that showed the role of music education on behavioral outcomes [69]. For example, Schellenberg et al. [40] showed that among Canadian children in 3rd and 4th grade (n = 38) who received music lessons for 10 months, those who initially exhibited low levels of prosocial skills showed

**Table 5. GEE results for the behavioral measures (aggression – PARB).**

| Variables in the model | B | SD. Error | 95% CI Lower bound | 95% CI Upper bound | Sig. |
|---|---|---|---|---|---|
| Intercept (outcome: Direct Aggression) | 13.691 | 0.9754 | 11.775 | 15.607 | <0.001 |
| Main effect | | | | | |
| Group (ref. control) | −1.320 | 0.5399 | −2.378 | −0.261 | 0.015 |
| Time (ref. baseline) | 0.537 | 0.5247 | −0.499 | 1.573 | 0.307 |
| Maternal education (ref. illiterate) | | | | | |
| High school (not complete) | −0.578 | 1.1105 | −2.760 | 1.604 | 0.603 |
| Undergraduate (not complete) | 0.046 | 0.9415 | −1.804 | 1.896 | 0.961 |
| Undergraduate (complete) | −0.637 | 1.0079 | −2.617 | 1.343 | 0.527 |
| Interaction (treatment effect) | | | | | |
| Group x time | 0.620 | 0.6637 | −0.685 | 1.924 | 0.351 |
| Scale | 5.956 | 0.2928 | | | |
| Intercept (outcome: Reactive Aggression) | 16.934 | 1.3996 | 14.191 | 19.678 | <0.001 |
| Main effect | | | | | |
| Group (ref. control) | −2.449 | 0.9092 | −4.231 | −0.667 | 0.007 |
| Time (ref. baseline) | 1.050 | 0.7708 | −0.473 | 2.572 | 0.175 |
| Maternal education (ref. illiterate) | | | | | |
| High school (not complete) | −1.754 | 1.5178 | −4.730 | 1.222 | 0.248 |
| Undergraduate (not complete) | −0.153 | 1.3028 | −2.708 | 2.403 | 0.907 |
| Undergraduate (complete) | −1.880 | 1.4553 | −4.732 | 0.973 | 0.197 |
| Interaction (treatment effect) | | | | | |
| Group x time | 0.834 | 0.9124 | −0.962 | 2.630 | 0.361 |
| Scale | 13.616 | 0.5929 | | | |
| Intercept (outcome: Internalizing Behavior) | 4.890 | 1.1156 | 2.697 | 7.083 | <0.001 |
| Main effect | | | | | |
| Group (ref. control) | −0.916 | 0.7003 | −2.289 | 0.457 | 0.191 |
| Time (ref. baseline) | 0.111 | 0.6458 | −1.161 | 1.383 | 0.864 |
| Maternal education (ref. illiterate) | | | | | |
| High school (not complete) | 0.661 | 1.1132 | −1.524 | 2.846 | 0.553 |
| Undergraduate (not complete) | 0.528 | 1.0367 | −1.510 | 2.566 | 0.611 |
| Undergraduate (complete) | −0.155 | 1.1420 | −2.403 | 2.094 | 0.892 |
| Interaction (treatment effect) | | | | | |
| Group x time | 0.060 | 0.7839 | −1.478 | 1.598 | 0.939 |
| Scale | 9.495 | 0.3629 | | | |
| Intercept (outcome: Teacher Support) | 3.162 | 0.8923 | 1.412 | 4.911 | <0.001 |
| Main effect | | | | | |
| Group (ref. control) | 0.709 | 0.6947 | −0.653 | 2.071 | 0.307 |
| Time (ref. baseline) | −0.419 | 0.6070 | −1.613 | 0.775 | 0.490 |
| Maternal education (ref. illiterate) | | | | | |
| High school (not complete) | −1.079 | 0.8845 | −2.813 | 0.655 | 0.223 |
| Undergraduate (not complete) | −0.795 | 0.7925 | −2.350 | 0.760 | 0.316 |
| Undergraduate (complete) | −0.295 | 0.9007 | −2.062 | 1.471 | 0.743 |
| Interaction (treatment effect) | | | | | |
| Group x time | −0.310 | 0.7526 | −1.787 | 1.167 | 0.680 |
| Scale | 8.026 | 0.4110 | | | |

**Table 6. GEE results for cognitive measures.**

| Variables in the model | B | SD. Error | 95% CI Lower bound | 95% CI Upper bound | Sig. |
|---|---|---|---|---|---|
| Intercept (outcome: Working Memory) | 1.671 | 0.3646 | 0.956 | 2.385 | <0.001 |
| Main effect | | | | | |
| Group (ref. control) | −0.454 | 0.2562 | −0.956 | 0.048 | 0.077 |
| Time (ref. baseline) | 0.595 | 0.2137 | 0.176 | 1.014 | 0.005 |
| Maternal education (ref. illiterate) | | | | | |
| High school (not complete) | 0.257 | 0.3526 | −0.434 | 0.948 | 0.467 |
| Undergraduate (not complete) | 0.514 | 0.3407 | −0.153 | 1.182 | 0.131 |
| Undergraduate (complete) | 0.297 | 0.3797 | −0.447 | 1.041 | 0.434 |
| Interaction (treatment effect) | | | | | |
| Group x time | 0.179 | 0.2644 | −0.340 | 0.698 | 0.498 |
| Scale | 1.135 | 0.0212 | | | |
| Intercept (outcome: Sustained Attention) | 38.228 | 9.5735 | 19.185 | 57.272 | <0.001 |
| Main effect | | | | | |
| Group (ref. control) | −9.413 | 6.8719 | −22.888 | 4.062 | 0.171 |
| Time (ref. baseline) | 13.156 | 5.5694 | 2.162 | 24.150 | 0.019 |
| Maternal education (ref. illiterate) | | | | | |
| High school (not complete) | −10.238 | 9.0796 | −28.246 | 7.771 | 0.262 |
| Undergraduate (not complete) | −6.081 | 7.4015 | −20.880 | 8.717 | 0.414 |
| Undergraduate (complete) | −8.131 | 10.4606 | −29.071 | 12.808 | 0.440 |
| Interaction (treatment effect) | | | | | |
| Group x time | −0.296 | 6.9673 | −14.032 | 13.440 | 0.966 |
| Scale | 566.821 | 18.6927 | | | |
| Intercept (outcome: Divided Attention) | 3.136 | 15.0566 | −26.380 | 32.653 | 0.835 |
| Main effect | | | | | |
| Group (ref. control) | −5.470 | 8.9511 | −23.014 | 12.074 | 0.541 |
| Time (ref. baseline) | 17.416 | 7.7807 | 1.999 | 32.833 | 0.027 |
| Maternal education (ref. illiterate) | | | | | |
| High school (not complete) | 10.380 | 14.1570 | −17.374 | 38.134 | 0.463 |
| Undergraduate (not complete) | 11.349 | 13.2503 | −14.625 | 37.323 | 0.392 |
| Undergraduate (complete) | 14.316 | 15.3329 | −15.750 | 44.381 | 0.351 |
| Interaction (treatment effect) | | | | | |
| Group x time | −0.854 | 8.8633 | −18.243 | 16.535 | 0.923 |
| Scale | 1401.124 | 55.1266 | | | |
| Intercept (outcome: Alternating Attention) | 22.372 | 8.0451 | 6.601 | 38.142 | 0.005 |
| Main effect | | | | | |
| Group (ref. control) | −2.072 | 7.1889 | −16.162 | 12.018 | 0.773 |
| **Time (ref. baseline)** | **19.482** | **5.6685** | **8.361** | **30.603** | **0.001** |
| Maternal education (ref. illiterate) | | | | | |
| High school (not complete) | 0.343 | 8.0772 | −15.500 | 16.186 | 0.966 |
| Undergraduate (not complete) | 5.607 | 5.3026 | −4.793 | 16.008 | 0.290 |
| Undergraduate (complete) | 7.997 | 6.8758 | −5.485 | 21.479 | 0.245 |
| Interaction (treatment effect) | | | | | |
| Group x time | −4.103 | 6.0616 | −15.997 | 7.790 | 0.499 |
| Scale | 520.552 | 20.2728 | | | |

improvements at FUP, compared to a control group (n = 46). Other authors showed that music training impacts children's mood, emotional regulation, and empathy [70].

Conversely, our study agrees with Villanueva et al.'s study [45] in which long-term after-school music training (4 years) did not impact socioemotional skills (measured by a sharing task) among American children (5–8 years; n = 84). In fact, children from an active control group (sport lessons) gave away stickers about 9% more than the music group each year. Similarly, Habibi et al. [71] did not prove music training effect among 6–7 years old American children (n = 45) who attended music lessons, in relation to controls (TAU and sport lessons), in a socioemotional measure (helping and sharing). Moreover, music classes did not impact hyperactivity and impulsivity symptoms of Italian children (*n* = 130; 8–10 years) measured by the SDAI rating scale evaluated by teachers, although the study showed an effect in the self-attributed symptoms [72].

Although previous studies [67] have found that higher maternal education levels (*i.e.*, at least some college education) are associated with fewer behavioral problems in children, maternal education lost significance in the GEE models. Given the quasi-experimental nature of our study, there was a clear selection bias for this variable. We hypothesize that the strong imbalance of mother education among the groups is influenced by schooling itself. In Brazil, children typically start school at the age of 6, with some beginning at age 7 (universal enrollment). Admission to the first year in public schools in Brazil occurs by March 31 of the year in which enrollment occurs for children who turn 6 years old. It means that some children start school at age 7. In our sample, 96% of the children aged 6–7. However, before this age, access to daycare and kindergarten is more limited for families with lower socioeconomic status (SES). Mothers with higher education levels may be more likely to enroll their children in programs like Guri, to avoid that their children stay alone at home in vulnerable areas such as the ones from which the sample was recruited.

In terms of cognitive outcomes, alternating attention had a developmental effect, increasing equally between groups, from baseline to FUP. A possible explanation for this finding is that in formal educational settings, children are more able to learn and retain new information, keeping up with other children's skills. During the pandemic, many children spent long periods of time in front of screens (*e.g.*, TV, tablets, games, and cell phones), which is known to reduce their attention span [73]. Contrary to earlier studies, we did not find effects of music instruction on children's attention, possibly due to our small sample size, the short duration of the intervention (*i.e.*, 7 months) and music classes (*i.e.*, only one hour per week), and the unprecedented pandemic variable.

Finally, another developmental effect was a reduction in overall difficulty. It's possible that, even among children with behavioral problems, life experiences with peers and the community can promote emotional regulation and coping with more complex situations, which is expected at this age group [74].

## Conclusion & Limitations

Parents of the children selected for this study reported high difficulties in the SDQ, which suggests a risk of exposure to mental health vulnerabilities. The surveys were carried out in peripheral regions of São Paulo, Brazil, where services such as public security, health, education, and transportation are scarce and oftentimes precarious where they exist. The literature indicates such elements as predictors of emotional symptoms, health problems related to emotional stress, and behavioral problems in children [75].

As a limitation of this study, the absence of statistically significant time x group interactions may be attributed to the small sample size and short intervention time. A major challenge was achieving participant adherence, which affected both groups. As a result, we were unable to achieve intended 2:1 ratio between the control and experimental groups. Moreover, since the study was not designed as a randomized trial, significant differences between the groups were identified, requiring adjustments for maternal education. This further reduced the effective sample size. Future studies should focus either on non-naturalistic music programs to enable randomization, or on naturalistic programs with entry criteria that do not conflict with ethical considerations, as seen with the Guri program.

Additionally, a question remains unanswered: Would different curricular approaches to music education yield different results? Previous studies suggest that curated curricula may yield more substantial findings in terms of both EFs [76] and socioemotional skills [77]. Additionally, the length of programs is important; the literature points to the potential of long-term exposure to music practice (more than two years) on the impact of behavioral measures and brain structures [3,8,78–80]. Our study collected data over a school year, which, in practice, had an interval between the baseline and FUP of only about seven months, and included an hour of music class per week, and in the aftermath of a global pandemic. We hope that the tendency observed here can contribute to future research. More studies are obviously needed.

## Supporting information

**S1 Table. TREND.** TREND statement checklist.
(PDF)

**S1 File. Trial Study Protocol Translation.** The effects on brain Structures, social and cognitive skills in children exposed to the Guri Santa Marcelina program in Greater São Paulo: A quasi-experimental study.
(PDF)

**S2 File. Trial Study Protocol.** Os efeitos sobre estruturas cerebrais, habilidades sociais e cognitivas em crianças expostas ao programa Guri Santa Marcelina na Grande São Paulo: Um estudo quasi-experimental.
(PDF)

## Acknowledgments

We thank all the families and their children who volunteered in this research, as well as the assistants who collected the data, the Projeto Guri (OS Santa Marcelina), Secretaria Municipal de Educação de São Paulo, and Secretaria da Cultura e Economia Criativa do Estado de São Paulo for granting permission for the study to take place.

## Author contributions

**Conceptualization:** Graziela Bortz, Andrea Parolin Jackowski, Hugo Cogo-Moreira, Patrícia Silva Lúcio.

**Data curation:** Graziela Bortz, Andrea Parolin Jackowski, Hugo Cogo-Moreira, Patrícia Silva Lúcio.

**Formal analysis:** Hugo Cogo-Moreira, Patrícia Silva Lúcio.

**Funding acquisition:** Graziela Bortz, Andrea Parolin Jackowski.

**Investigation:** Graziela Bortz, Andrea Parolin Jackowski.

**Methodology:** Andrea Parolin Jackowski, Hugo Cogo-Moreira, Patrícia Silva Lúcio.

**Project administration:** Graziela Bortz.

**Resources:** Graziela Bortz, Andrea Parolin Jackowski.

**Software:** Andrea Parolin Jackowski, Patrícia Silva Lúcio.

**Supervision:** Graziela Bortz, Patrícia Silva Lúcio.

**Validation:** Beatriz Ilari, Nayana Di Giuseppe Germano, Andrea Parolin Jackowski, Hugo Cogo-Moreira, Patrícia Silva Lúcio.

**Visualization:** Beatriz Ilari, Nayana Di Giuseppe Germano, Hugo Cogo-Moreira.

**Writing – original draft:** Graziela Bortz.

**Writing – review & editing:** Graziela Bortz, Beatriz Ilari, Nayana Di Giuseppe Germano, Andrea Parolin Jackowski, Hugo Cogo-Moreira, Patrícia Silva Lúcio.

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
