## [Decision Letter · Decision Letter 0]

22 Dec 2024

Dear Dr. Bortz,

Thank you for submitting your manuscript to PLOS ONE. After careful consideration, we feel that it has merit but does not fully meet PLOS ONE’s publication criteria as it currently stands. Therefore, we invite you to submit a revised version of the manuscript that addresses the points raised during the review process.

Please submit your revised manuscript by Feb 05 2025 11:59PM. If you will need more time than this to complete your revisions, please reply to this message or contact the journal office at plosone@plos.org . A rebuttal letter that responds to each point raised by the academic editor and reviewer(s). You should upload this letter as a separate file labeled 'Response to Reviewers'.A marked-up copy of your manuscript that highlights changes made to the original version. You should upload this as a separate file labeled 'Revised Manuscript with Track Changes'.An unmarked version of your revised paper without tracked changes. You should upload this as a separate file labeled 'Manuscript'.

We look forward to receiving your revised manuscript.

Kind regards,

Mu-Hong Chen, M.D., Ph.D.

Academic Editor

PLOS ONE

Journal Requirements:

“Fapesp Public Policies Research Program (Proc. n. 2019/02133-4).

Fundunesp (Proc. n. 3334/2022 – PIF).”

4. We note that you have referenced (unpublished) on page 6, which has currently not yet been accepted for publication. Please remove this from your References and amend this to state in the body of your manuscript: (ie “Bewick et al. [Unpublished]”) as detailed online in our guide for authors

5. We note that the original protocol file you uploaded contains a confidentiality notice indicating that the protocol may not be shared publicly or be published. Please note, however, that the PLOS Editorial Policy requires that the original protocol be published alongside your manuscript in the event of acceptance. Please note that should your paper be accepted, all content including the protocol will be published under the Creative Commons Attribution (CC BY) 4.0 license, which means that it will be freely available online, and any third party is permitted to access, download, copy, distribute, and use these materials in any way, even commercially, with proper attribution.

Therefore, we ask that you please seek permission from the study sponsor or body imposing the restriction on sharing this document to publish this protocol under CC BY 4.0 if your work is accepted. We kindly ask that you upload a formal statement signed by an institutional representative clarifying whether you will be able to comply with this policy. Additionally, please upload a clean copy of the protocol with the confidentiality notice (and any copyrighted institutional logos or signatures) removed.

Reviewers' comments:

Reviewer's Responses to Questions

**Comments to the Author**

1. Is the manuscript technically sound, and do the data support the conclusions?

Reviewer #1: Partly

Reviewer #2: No

2. Has the statistical analysis been performed appropriately and rigorously?

Reviewer #1: I Don't Know

Reviewer #2: No

3. Have the authors made all data underlying the findings in their manuscript fully available?

Reviewer #1: Yes

Reviewer #2: No

4. Is the manuscript presented in an intelligible fashion and written in standard English?

Reviewer #1: No

Reviewer #2: Yes

Reviewer #1: Important note: This review pertains only to ‘statistical aspects’ of the study and so ‘clinical aspects’ [like medical importance, relevance of the study, ‘clinical significance and implication(s)’ of the whole study, etc.] are to be evaluated [should be assessed] separately/independently. Further please note that any ‘statistical review’ is generally done under the assumption that study specific methodological [as well as execution] issues are perfectly taken care of by the investigator(s). This review is not an exception to that and so does not cover clinical aspects {however, seldom comments are made only if those issues are intimately / scientifically related & intermingle with ‘statistical aspects’ of the study}. Agreed that ‘statistical methods’ are used as just tools here, however, they are vital part of methodology [and so should be given due importance]. I look at the manuscript in/with statistical view point, other reviewer(s) look(s) at it with different angle so that in totality the review is very comprehensive. However, there should be efforts from authors side to improve (may be by taking clues from reviewer’s comments). Therefore, please do not limit the revision only (with respect) to comments made here.

COMMENTS: I have different opinion/views/observations/concerns or rather questions regarding quite a few issues [including some very serious ones] which are given below:

I noted that your ABSTRACT is drafted alright (in my opinion), but is ‘assay type’. It is preferable [refer to item 1b of CONSORT checklist 2010: Structured summary of trial design, methods, results, and conclusions] to divide the ABSTRACT with small sections like ‘Objective(s)’, ‘Methods’, ‘Results’, ‘Conclusions’, etc. which is an accepted practice of most of the good/standard journals [including this one, though ‘The PLoS One Guidelines to Authors’ did not specify an Abstract format, it is desirable]. It will definitely be more informative then, I guess, whatever the article type may be {though Section headings may differ for different Article Types [example: Study Protocol]}. Moreover, your abstract is very short (only 188 words).

Note that the effect size {eta-squared from an analysis of variance (ANOVA) test} [A significant effect of the intervention in emotional symptoms in the music group as opposed to controls was found (F(1,89) = 4.562, p = 0.035, �2 = 0.049), indicating benefits for the music group] is very-very small. Reference quoted in line 326 {13. Miller E, Cohen J. An Integrative Theory of Prefrontal Cortex Function. Annual review of neuroscience. 2001; 24: 167-202 is different } is diffent than one given below. How that indicates “benefits for the music group” is a serious question. The manuscript is on very valid & very interesting topic. However, in such studies/trials the effect size generally observed/expected to be small and so needs ‘large sample’ sizes (for both groups). Please refer to table-1 on page 157 of Jacob Cohen’s paper “A power primer” in Psychological Bulletin, 1992, vol.:112, pp 155-159 [which is a sort of summary of the excellent book by Cohen himself titled ‘Statistical power analysis for the behavioral sciences’, Academic Press, 1977, New York] for magnitude of various effect size indexes. How the ‘effect of the intervention in emotional symptoms’ measured (as one entity) is not described (most questionable)?

Inference “Children whose mothers had low levels of education benefited with gains between 0.70 and 0.95 standard deviations in both groups for divided attention, indicating a significant interaction with maternal education” is wonderful [least understood, more explanation required]. From where or how it is drawn is not understood (highly questionable)? This way of interpreting/expressing result(s) appears to be new to me. Kindly let me learn more/novel things {by giving details}.

Most objectional/questionable issue is regarding ‘sample size’. Footnote-4 of line 169 (we intended to test children aged between 6 and 7, but as it was difficult to achieve the minimum sample size desired in this group, the range was extended) shows that the formal ‘desired/required sample size’ estimation was done sometime. But surprisingly it is not discussed at all (anywhere) in the manuscript. You have not given/discussed ‘How the required minimum sample size for this study was determined’ which nevertheless is a very-very important question [one of the important items in CONSORT checklist, item 7a] for any type of study (clinical trial or else). This point needs to be discussed in adequate details {including assumptions made at the time of estimation, power (confidence/accuracy/precision in case of single-arm/group studies) of the study, software used, etc.}. One concludes that the sample size used in this study is haphazardly determined. {even for required sample size use the above reference [“A power primer” in Psychological Bulletin, 1992, vol.:112, pp 155-159]}

As you may (rather definitely) know that “all ‘Clinical Trials’ must follow CONSORT guidelines”. Since your article type is ‘Clinical Trial’, you are supposed to cover these items in the report. In addition, you may please refer to “ICH HARMONISED TRIPARTITE GUIDELINE STATISTICAL PRINCIPLES FOR CLINICAL TRIALS - E9” latest version of which is available on NET/WWW. Although you mentioned in the title itself [and also clarified later in lines 163-66: Due to ethical reasons related to the policies of enrollment in the Guri Program, it was not possible to randomize child participants into music and control groups. Therefore, we performed an intervention study based on a quasi-experimental design] that it ‘A quasi-experimental study’, the journal rightly classified this study as a clinical trial. Choice/selection of the ‘control’ group is doubtful regarding its correctness. Please note that non-random allocation does not make it a different (like observational) study and so you are supposed to follow “CONSORT guidelines”.

Though the measures/tools used are appropriate often times, most of them [example: statement in lines 187-88 After signing consent forms, parents were invited to complete the SDQ–parent version and the ABEP questionnaire] are likely to yield data that are in ‘ordinal’ level of measurement [and not in ratio level of measurement for sure {as the score two times higher does not indicate presence of that parameter/phenomenon as double (for example, a Visual Analogue Scales VAS score or say ‘depression’ score)}]. In such situation one has to inflate sample size by 10 to 20% and also then the application of suitable non-parametric (or distribution free) test(s) is/are indicated/advisable [even if distribution may be ‘Gaussian’ (also called ‘normal’)]. Agreed that there is/are no non-parametric test(s)/technique(s) available to be used as alternative in all situation(s), but should be used whenever/wherever they are available. Therefore, in short use suitable non-parametric test(s)/technique(s) while dealing with data that are in ‘ordinal’ level of measurement even if [despite that] the distribution may be ‘Gaussian’. Testing ‘normality’ in sample [by using any normality test(s)} is not required/desired while dealing with data that are in ‘ordinal’ level of measurement [as most of the normality tests are not valid for ‘ordinal’ data].

In the ABSTRACT [The music group (n = 38, 5-9 years) was recruited from 10 centers (polos) distributed across the metropolitan area of São Paulo and the control group (n = 67) consisted of aged-matched children who attended public schools surrounding the polos.] as well as in lines 181-2 it is stated that “The control group consisted of aged-matched children who attended regular schools surrounding the Guri polos” but sample sizes of these two groups are different, therefore I am curious to know the method used here for matching. Please note that similar age group coverage does not mean ‘aged-matched’. What exactly you want to convey by saying in lines 315-6 “We paired the groups using pre-test data (before running the analyses, using data from all children who were paired).”. In my knowledge, the term is ‘age-matched’ and not ‘aged-matched’. Kindly check for the ‘English’ language. Agreed that English is not our mother tongue (definitely not mine, may or may not be yours but certainly not of many readers), however in any case, remember/ Kindly mind you [please excuse me for such a harsh comment/statement] that this is a scientific/academic document and so all details should be clearly/correctly communicated (do not take reader’s for granted). You may take help of language professional expert, if needed.

Way used for matching (lines 288-290: “we managed the sample characteristics to match study groups. Sample matching was carried out based on the removal of children from both groups”) is also highly objectionable. Difference (large) in sample sizes of two groups [experimental/intervention group & control group] is also questionable. Method of imputation used [an expectation–maximization (EM) algorithm which is an iterative method to find (local) maximum likelihood or maximum a posteriori (MAP) estimates of parameters in statistical models, where the model depends on unobserved latent variables] is also questionable. Application of expectation–maximization (EM) algorithm as a method of imputation is not found on NET/WWW search {above quote about this method is after/by NET/WWW search only}.

When only two independent groups are to be compared {ex. Women and Men}, we use ‘t’ test for two independent groups [non-parametric equivalent to unpaired ‘t’ test is Mann-Whitney ‘U’ test] and not ANOVA ‘F’. Although ‘F’ and ‘t’ are mathematically related/equivalent [square of ‘t’ is exactly equal to ‘F’ if (mistakenly) calculated for two groups], logic/philosophy (and so underlying assumptions) behind their development and algorithms used for estimation of test statistic are different. They are applicable in different situations. I do not see any ‘repeated measures’ (lines 320-321: Inferential analyses are reported by ANOVAS with repeated measures). ‘Pre-Post’ differences could be delt by Mann-Whitney test on ‘change scores’ or by applying ANCOVAs. In fact, whole ‘Statistical analysis’ (including section described in lines 310 onwards) is of questionable value.

Limitation of this study highlighted in lines 484-5 “As limitations of this study, the absence of statistically significant interaction between the groups in some cases may be due to the low sample size and short intervention time” is appreciated but note that there are quite a few other limitations as well [important ones are ‘design’ of the study and way of analyses].

As pointed out in ‘important note’ above “This review pertains only to ‘statistical aspects’ of the study and so ‘clinical aspects’ should be assessed separately/independently [one should carefully consider/look at the clinical implications of the study]. In my opinion, to rescue this article (which little difficult, but not impossible), large amount of re-vision (re-drafting) may be needed. However, please do not limit the revision only (with respect) to comments made here. More improvement is expected. Nevertheless, ‘how to handle/accommodate these suggestions?’ is questionable as the study is already conducted/complete. ‘Major revision’ is recommended [as the study is on very interesting topic and therefore, would like to give chance to authors for improvement of the manuscript].

Reviewer #2: In Table 3, the main effect of time on emotional symptoms is not statistically significant. Although a significant group × time interaction is observed, this finding alone does not conclusively demonstrate that the music intervention effectively reduces emotional symptoms in the intervention group—an issue of paramount clinical importance. Therefore, additional statistical analyses are recommended to more definitively assess the intervention’s impact within each group.

**Do you want your identity to be public for this peer review?** For information about this choice, including consent withdrawal, please see our Privacy Policy

Reviewer #1: No

Reviewer #2: No

---

## [Author Response · Author response to Decision Letter 1]

15 Jun 2025

To both reviewers #1 and #2:

We are very grateful for the feedback and observations regarding the CONSORT guidelines. We realized that a different approach to reporting the study was necessary. Alongside this letter and the revised manuscript, we have carefully addressed all the questions and observations raised through the reviewers’ attentive reading of our work.

Because the revised manuscript underwent significant changes—particularly due to modifications in the statistical analysis—we took the liberty of reporting all the changes directly in the text, rather than responding point by point. All modifications are highlighted in red in the manuscript.

- In accordance with item 1b of the CONSORT 2010 checklist, we revised the structure of the Abstract.

- As noted by Reviewer 1, our study is best described as a quasi-experimental study, and the journal has correctly classified it as a clinical trial. Therefore, it must adhere to the CONSORT guidelines. We acknowledge a serious mistake in our sample matching and imputation methods. According to CONSORT, the complete dataset of the selected sample must be reported, and Little’s MCAR test is not the most appropriate choice for this type of data. Based on this observation, we returned to the protocol and revised our inclusion criteria.

- In the Methods section, we added a Design subsection to clarify the quasi-experimental nature of the study. In the Sample Recruitment section, we provided a more detailed explanation of the target sample (see page 8 onwards) and the exclusion process prior to sample assignment. We also added Footnote 4 to explain the recruitment criteria and amendments to the study protocol. Consequently, the flowchart (Fig. 1) was updated. All supplementary tables and Figure 1 have been modified, and supplementary tables as well as Figures 2 through 4 have been deleted.

- For the statistical analysis, multiple imputation was used instead of Little’s MCAR test. We conducted independent samples t-tests and chi-squared tests to examine baseline differences between the music group and the control group. Since SPSS does not provide pooled results for multiply imputed data, standard deviations, Cohen’s d, and chi-square statistics were manually calculated in Excel using Rubin’s rules [Rubin DB. Multiple imputation for nonresponse in surveys. Wiley; 1987].

- To test the intervention effect, generalized linear models (GLMs) were used instead of ANOVAs, as they are more appropriate for this type of data. Maternal education was included as a covariate due to baseline differences between the groups. Finally, Bonferroni’s correction was applied to adjust p-values for multiple comparisons.

- As a result, our data did not support significant differences between the groups in terms of behavioral problems, cognitive abilities, or aggression. The few observed effects point to developmental changes in both groups. For this reason, we also changed the title to: The impact of music education on children's cognitive and socioemotional development: A quasi-experimental study in the Guri Program in Brazil.

Although the results—and consequently their interpretation—have changed substantially, we still believe the paper makes a meaningful contribution to the literature on music education and its potential impact on behavioral characteristics. Few clinical trials have explored the relationship between music education, cognitive and socioemotional outcomes. The methodological changes introduced aim to improve the assessment of this relationship, and implications for future research are discussed.

---

## [Decision Letter · Decision Letter 1]

8 Jul 2025

Dear Dr. Graziela Bortz,

Thank you for submitting your manuscript to PLOS ONE. After careful consideration, we feel that it has merit but does not fully meet PLOS ONE’s publication criteria as it currently stands. Therefore, we invite you to submit a revised version of the manuscript that addresses the points raised during the review process.

**ACADEMIC EDITOR: **

We look forward to receiving your revised manuscript.

Kind regards,

Mu-Hong Chen, M.D., Ph.D.

Academic Editor

PLOS ONE

Reviewers' comments:

Reviewer's Responses to Questions

**Comments to the Author**

Reviewer #1: All comments have been addressed

Reviewer #2: All comments have been addressed

2. Is the manuscript technically sound, and do the data support the conclusions?

Reviewer #1: (No Response)

Reviewer #2: No

3. Has the statistical analysis been performed appropriately and rigorously?

Reviewer #1: (No Response)

Reviewer #2: No

4. Have the authors made all data underlying the findings in their manuscript fully available?

Reviewer #1: (No Response)

Reviewer #2: Yes

5. Is the manuscript presented in an intelligible fashion and written in standard English?

Reviewer #1: (No Response)

Reviewer #2: Yes

Reviewer #1: COMMENTS: I note that most of the comments made on earlier draft are attended to the possible extend. However, the original over-all quality of the manuscript generally remains the same [as the study is already completed & I am not fully satisfied]. Now I recommend the acceptance of this manuscript for publication [only because the study (is on important topic and) has potential]. However, note the following:

Although all modifications are highlighted in red in the manuscript, reporting the changes directly in the text, makes the review process difficult and time consuming. Responding to comments point by point could have been very useful and so appreciated.

Reviewer #2: I appreciate the authors’ thoughtful and extensive revisions to the manuscript. However, the revised statistical approach still does not appropriately test the study’s central hypothesis.

The current modeling strategy uses follow-up scores as the dependent variable while adjusting for baseline and maternal education. This design may control for baseline differences but does not directly test whether the groups change differently over time.

Without modeling the group × time interaction or using a repeated-measures framework, the analyses do not support valid inferences about the effectiveness of the music education intervention.

I strongly encourage the authors to reanalyze the data using a repeated-measures GLM that includes time, group, and their interaction. This is standard practice for evaluating intervention effects in longitudinal designs.

**Do you want your identity to be public for this peer review?** For information about this choice, including consent withdrawal, please see our Privacy Policy

Reviewer #1: **Yes: ** Dr. Sanjeev Sarmukaddam

Reviewer #2: No

---

## [Author Response · Author response to Decision Letter 2]

21 Aug 2025

Dear editor and reviewers,

We are immensely grateful for the careful reading and comments received, which certainly improved the quality of our manuscript. We would like to inform you that all the reviewers' suggestions were considered in this revision of the manuscript, as follows:

Reviewer #1: COMMENTS: I note that most of the comments made on earlier draft are attended to the possible extend. However, the original over-all quality of the manuscript generally remains the same [as the study is already completed & I am not fully satisfied]. Now I recommend the acceptance of this manuscript for publication [only because the study (is on important topic and) has potential]. However, note the following:

Although all modifications are highlighted in red in the manuscript, reporting the changes directly in the text, makes the review process difficult and time consuming. Responding to comments point by point could have been very useful and so appreciated.

We greatly appreciate your consideration in accepting the manuscript, even though you likely identified methodological or technical issues with the study's execution.

We regret that our response method made it difficult for you to understand the changes. Because the statistical analysis changed drastically from Round 1 to Round 2, and consequently the results and their interpretation, we chose to summarize the changes in the letter. We will consider this comment in future submissions, and we are grateful for it.

Reviewer #2: I appreciate the authors’ thoughtful and extensive revisions to the manuscript. However, the revised statistical approach still does not appropriately test the study’s central hypothesis.

The current modeling strategy uses follow-up scores as the dependent variable while adjusting for baseline and maternal education. This design may control for baseline differences but does not directly test whether the groups change differently over time.

Without modeling the group × time interaction or using a repeated-measures framework, the analyses do not support valid inferences about the effectiveness of the music education intervention.

I strongly encourage the authors to reanalyze the data using a repeated-measures GLM that includes time, group, and their interaction. This is standard practice for evaluating intervention effects in longitudinal designs.

We greatly appreciate your feedback. We have reorganized the database and modified the statistical analysis. We use Generalized Estimating Equations (GEE) instead of General Linear Models (GLM), following this recommendation. On pages 12-13, lines 295-302 (Statistical Analysis), we state:

The intervention effect was analyzed using Generalized Estimating Equations (GEE), an approach suitable for modeling longitudinal or repeated-measures data, as it considers the dependence of observations within individuals and groups. Respondents were specified as the within-subject factor, and assessment time (baseline and FUP) was modeled as the repeated measure. An exchangeable working correlation structure was assumed. Group and assessment time were included as fixed factors, with maternal education entered as a covariate. Main effects were reported, and the group × time interaction was tested to assess the intervention’s effectiveness.

Moreover, in the results section, we explained how to interpret the results accordingly (page 16, lines 344-359):

GEE models were used to verify the impact of musical intervention on the different variables of the study. For the SDQ analysis, difficulties (i.e., hyperactivity, relationship problems, conduct problems, emotional symptoms, and difficulties in general) and prosocial behavior were considered as dependent variables. We considered the group (control, experimental) and time (baseline, FUP) as factors and maternal education as covariate. Due to the ordinal nature of this variable, dummy codes were created using the lowest level of the maternal education (illiterate) as the reference variable. The control group and baseline were used as references.

In the GEE analysis, the main effect of groups indicates whether the groups differ significantly on the target variable, regardless of time. Similarly, the main effect of time indicates whether scores on the target variable change (decrease or increase) regardless of group. These results are adjusted for maternal education. The effect of the intervention is reflected in the presence of a significant group × time interaction, which shows whether changes in the dependent variable (decrease or increase) differ between groups. We expect a greater reduction in symptoms or a greater increase in cognitive variables for the treatment group compared to the control group. Baseline and FUP comparisons are shown in Table 4.

Some results were changed with the correct analysis. In this version of the manuscript, no significant results were found, except for changes in developmental difficulties (total score) and alternating attention.

We appreciate the time you took to carefully review our work.

---

## [Decision Letter · Decision Letter 2]

29 Sep 2025

The impact of music education on children's cognitive and socioemotional development: A quasi-experimental study in the Guri Program in Brazil

PONE-D-24-48345R2

Dear Dr. Graziela Bortz,

We’re pleased to inform you that your manuscript has been judged scientifically suitable for publication and will be formally accepted for publication once it meets all outstanding technical requirements.

Kind regards,

Mu-Hong Chen, M.D., Ph.D.

Academic Editor

PLOS ONE

Additional Editor Comments (optional):

Reviewers' comments:

Reviewer's Responses to Questions

**Comments to the Author**

Reviewer #2: All comments have been addressed

2. Is the manuscript technically sound, and do the data support the conclusions?

Reviewer #2: Yes

3. Has the statistical analysis been performed appropriately and rigorously?

Reviewer #2: Yes

4. Have the authors made all data underlying the findings in their manuscript fully available?

Reviewer #2: Yes

5. Is the manuscript presented in an intelligible fashion and written in standard English?

Reviewer #2: Yes

Reviewer #2: The authors have addressed my previous concerns, particularly by implementing a repeated-measures framework (GEE) that includes group × time interactions. While the study is limited by a small sample size and a relatively short intervention period, the methods are sound, the analysis is now appropriate. Given PLOS ONE’s criteria of methodological rigor and clarity, I support the acceptance of this manuscript in its current form.

**Do you want your identity to be public for this peer review?** For information about this choice, including consent withdrawal, please see our Privacy Policy

Reviewer #2: No

---

## [Editor Report · Acceptance letter]

PONE-D-24-48345R2

PLOS ONE

Dear Dr. Bortz,

I'm pleased to inform you that your manuscript has been deemed suitable for publication in PLOS ONE. Congratulations! Your manuscript is now being handed over to our production team.

Kind regards,

on behalf of

Dr. Mu-Hong Chen

Academic Editor

PLOS ONE